# Exploring the Landscape of Breast Cancer Prevention among Chinese Residents in Italy: An In-Depth Analysis of Screening Adherence, Breast Self-Examination (BSE) Practices, the Role of Technological Tools, and Misconceptions Surrounding Risk Factors and Symptoms

**DOI:** 10.3390/ijerph21030308

**Published:** 2024-03-06

**Authors:** Luana Conte, Roberto Lupo, Serena Sciolti, Alessia Lezzi, Ivan Rubbi, Stefano Botti, Maicol Carvello, Annarita Fanizzi, Raffaella Massafra, Elsa Vitale, Giorgio De Nunzio

**Affiliations:** 1Laboratory of Biomedical Physics and Environment, Department of Mathematics and Physics “E. De Giorgi”, University of Salento, 73100 Lecce, Italy; luana.conte@unisalento.it (L.C.); giorgio.denunzio@unisalento.it (G.D.N.); 2Laboratory of Advanced Data Analysis for Medicine (ADAM), University of Salento, Local Health Authority, 73100 Lecce, Italy; 3“San Giuseppe da Copertino” Hospital, Local Health Authority, 73043 Lecce, Italy; robertolupo_2015@libero.it; 4Independent Researcher, 73100 Lecce, Italy; serenasciolti91@hotmail.it; 5National Cancer Association (ANT) Italia ONLUS Foundation, 73100 Lecce, Italy; alessia.lezzi@ant.it; 6School of Nursing, University of Bologna, 48018 Faenza, Italy; 7Haematology Unit, Azienda USL-IRCCS of Reggio Emilia, 42100 Reggio Emilia, Italy; stefano.botti@ausl.re.it; 8Ravenna Community Hospital, Local Health Authority Romagna, 48100 Ravenna, Italy; maicol.carvello2@unibo.it; 9Laboratorio di Biostatistica e Bioinformatica, IRCCS Istituto Tumori “Giovanni Paolo II”, 70126 Bari, Italy; a.fanizzi@oncologico.bari.it (A.F.); r.massafra@oncologico.bari.it (R.M.); 10Scientific Directorate, IRCCS Istituto Tumori “Giovanni Paolo II”, 70126 Bari, Italy; e.vitale@oncologico.bari.it

**Keywords:** Chinese residents in Italy, breast cancer screening, breast self-examination, prevention

## Abstract

Background: Breast cancer remains a significant health concern among women globally. Despite advancements in awareness and diagnostic techniques, it persists as a leading cause of death, with profound impacts on affected individuals’ quality of life. Primary and secondary prevention, including regular screenings and practices like breast self-examination (BSE), are pivotal in ensuring early diagnosis. The national health system (NHS) in Italy offers screenings for women aged 50–69 every two years, managed by the local health authority. However, the participation rates, especially among the Chinese female population residing in Italy, are not well understood. Methods: Using a snowball method, we electronically disseminated a survey to investigate how Chinese women living in Italy engage with available NHS screening programs. The survey also explores their practice of BSE and the use and impact of technological tools on prevention. Furthermore, the study aims to understand the subjects’ depth of knowledge and misconceptions about breast cancer. Results: The data reveal a significant gap in breast cancer screening adherence and knowledge among Chinese women in Italy, with a notable discrepancy between the general population and those who have previously encountered cancer. Conclusions: The results highlight the urgent need for interventions that are culturally sensitive, stressing that these actions are not only desirable but essential.

## 1. Introduction

Breast cancer continues to be the most common cancer globally [1,2]. As highlighted by GLOBOCAN 2020’s data on cancer diagnosis and mortality, breast cancer in women now stands as the most frequently diagnosed cancer and is the fifth leading cause of cancer-related deaths. Annually, it accounts for approximately 2.3 million new cases and 685,000 deaths [3]. In terms of cancer diagnoses in women, breast cancer constitutes 1 out of every 4 cases and is responsible for 1 in 6 of all cancer-related deaths. Globally, 1 in 18 women will face a breast cancer diagnosis during her lifetime [4].

Turning our attention to China, breast cancer profoundly affects its society. According to GLOBOCAN 2020, data from 23 Chinese cancer registries reveal that breast cancer ranks as the most common cancer among Chinese women. In 2020, there were 416,371 reported new cases and 117,174 deaths in a population of 1.411 billion. This represents 18% of global new cases and 17% of worldwide deaths for that year [5]. In addition, the frequency and fatality rate of breast cancer among Chinese women are on the rise, and the average age of diagnosis is relatively young. Studies suggest that the median age of diagnosis for breast cancer in Chinese women is in the late 40s to early 50s. This is younger when compared with Western countries, where the median age at diagnosis is typically in the early 60s [6,7,8,9]. This indicates the need for standardizing breast cancer screening in Chinese women [10].

In Italy, 55,700 new diagnoses and 12,500 deaths are expected in 2022 [11]. Even with heightened public consciousness, mortality due to breast cancer remains alarmingly high globally, with Italy reporting it as the primary cause of death [11]. In Italy, the estimated age-adjusted annual incidence of breast cancer was 94.2 per 100,000, with a mortality rate of 23.1 per 100,000 in 2012. In contrast, China’s estimated breast cancer incidence rate was 32.43 per 100,000, with a mortality rate of 8.65 per 100,000. This indicates a higher incidence and mortality rate of breast cancer in Italy compared with China and highlights the significant differences between the two countries in terms of breast cancer burden [12].

However, in terms of Chinese women residing in Italy, the scenario remains largely uncharted. Comprehensive studies and data detailing the incidence, diagnosis, and treatment outcomes for this specific group are limited, highlighting a gap in our understanding and emphasizing the need for targeted research in this area.

Early intervention is paramount when determining the appropriate treatment or surgical procedure for early-stage tumors. Enhanced diagnostic methods have enabled early detection, allowing for the identification of minuscule tumors that may not be easily detectable. Thus, early diagnosis remains the most vital factor in improving patient prognosis, with prevention and adherence to screening programs being the most effective strategies [13].

In Italy, consistent with preventative guidelines, biennial mammography screenings, offered by the national health system (NHS) through a local health authority (LHA), are freely accessible to women between the ages of 50–69 [11]. However, not everyone avails of this service, leading to stark regional differences in screening uptake, breast cancer rates, and survival outcomes [14,15]. A potential reason might be the insufficiency of knowledge about cancer risk factors, manifestations, and preventive tactics, as highlighted by a European survey pointing to the suboptimal awareness in Italy about routine screening tests [16].

An escalated risk of developing this cancer is notably associated with advancing age [11] and is intertwined with numerous determinants, including genetics, hormonal changes, diet, environment influences, lifestyle decisions, and prior breast conditions. Intriguingly, over half of these cases cannot be traced back to any known risk factor [17,18,19,20,21,22,23,24,25,26,27,28,29,30]. Mammograms are indispensable, particularly when initial breast cancer signs like detectable lumps emerge [31].

Another highly advocated preventive measure is breast self-examination (BSE). BSE entails women regularly checking their own breasts for any unusual alterations, which should then be communicated to their medical professional. While relying solely on self-examination might not suffice in a time where advanced tools like mammography and breast ultrasounds can pinpoint minute tumors, consistent and accurate BSE can indeed lower the chances of detecting late-stage breast cancer. Furthermore, this self-check is both practical and economical, and women are encouraged to adopt it from their early twenties [32]. Educating women about the benefits of BSE is crucial, as it is the easiest and the most straightforward method by which to detect breast cancer at an early stage [32,33]. Furthermore, with the advancement of technology, digital applications and tools propelling BSE and cancer awareness are becoming increasingly relevant.

Breast cancer remains a considerable public health issue, representing a leading cause of anxiety and premature mortality among women globally [34]. Given the ever-evolving landscape of healthcare, the emphasis on timely screening, meticulous self-examination for the earliest indicators, and the incorporation of technological advancements is undeniably pivotal for the early diagnosis of breast cancer. Within this broader context, the experiences and attitudes of specific communities become paramount, especially when considering the unique challenges they might face in accessing and understanding healthcare practices. One such community is that of Chinese women residing in Italy.

The focus on Chinese women living in Italy stems from an identified gap in the literature regarding this particular demographic’s health behaviors and access to healthcare services, especially in relation to breast cancer screening. Evidence suggests that the Chinese community in Italy has lower participation rates in organized screening programs, which raises concerns about potential health disparities.

Our study delves into the attitudes of Chinese women residing in Italy towards clinical screenings for breast cancer prevention, focusing on the 50–69 age group. This cohort, being recipients of free screening programs offered by the NHS, often faces cultural, linguistic, and occasionally socioeconomic challenges. These barriers might impact their participation, and the nuances of their adherence to such programs remain largely unknown. While prevention is undeniably crucial, an informed understanding of cancer stands equally vital. The more individuals grasp the risk factors and symptoms associated with breast cancer, the more proactive they tend to be towards screenings. In alignment with this understanding, our study seeks to understand how Chinese women in Italy engage with screening programs, how often they perform BSE, their level of awareness about breast cancer, and the influence of technology in supporting these habits. This understanding is vital to shaping future preventive strategies and interventions.

## 2. Materials and Methods

### 2.1. Study Design

From April 2023 to October 2023, we conducted a survey targeting Chinese women residing in Italy. In this study, we employed the same survey instrument that was previously utilized in our research detailed in [32], which examined the broader Italian population. Out of the individuals approached, 1144 chose to participate. The method of data collection involved an anonymous voluntary questionnaire. Inclusion criteria stipulated participants to be Chinese females, living in Italy, aged between 20 and 69 years who provided their informed consent to participate. Exclusions were made for those not fitting these criteria. The questionnaire was digitized using a predefined form on the Google Drive platform, and the survey was disseminated electronically. Various Facebook groups and Instagram pages were approached in order to circulate the digital questionnaire. The sampling technique employed was the (virtual) snowball method, which was continued until data saturation was achieved.

### 2.2. Survey Instrument

The questionnaire was constructed ‘ad hoc’ and was translated into Chinese by a native speaker. It consists of 81 items divided into 7 sections: the first section (7 items) gathers socio-demographic data, including age, geographical area of residence, marital status, level of education, employment status, and years of residence in Italy. The second section (25 items) assesses Chinese women’s access to Italian healthcare services, including questions about enrollment in the National Health Service (NHS) and its services. The third section (9 items) evaluates the participation of the women aged 50–69, in mammography screenings offered by the NHS, their frequency, and any reasons for non-participation. The fourth section (15 items) explores participants’ awareness and practice of BSE, as well as the frequency with which they perform it. The fifth section (5 items) delves into the use of technological tools, particularly smartphone apps, related to breast health and BSE practices. The sixth (10 items) and the seventh sections (10 items) focus on general knowledge and beliefs about breast cancer, encompassing causes, symptoms, and associated risk factors.

### 2.3. Ethical Considerations

The ethical considerations of the study were clearly outlined in the questionnaire introduction. The design of the questionnaire adhered to the principles set by the Italian Data Protection Authority (DPA). It was highlighted that participation was entirely voluntary, and participants could withdraw from the study at any time. Those expressing interest in participating received an informed consent form that reiterated the voluntary nature of participation and assured the confidentiality and anonymity of the collected data. All participants provided informed consent, underscoring the commitment to ethical research practices and participant rights. To further safeguard participants’ privacy, all questionnaire responses were de-identified.

### 2.4. Statistical Analysis

Data collected from the questionnaire responses of all of the participants were processed by descriptive statistical analysis. To highlight behaviors related to breast cancer, the participants were divided into two groups: Group A (*n* = 1118) represented the general population, while Group B (*n* = 26) consisted of women previously diagnosed or currently living with breast cancer. Continuous variables were articulated as mean and standard deviation (SD), while categorical variables were delineated using frequencies and percentages. The Mann–Whitney U-test was employed to discern differences between the two groups. A *p*-value less than 0.05 was deemed statistically significant. MATLAB software R2023B update 6 was utilized for all statistical computations, encompassing both qualitative and quantitative data assessments.

## 3. Results

The baseline characteristics were evaluated for all respondents and data were collected. In order to ascertain the possible difference in women’s information, knowledge, and beliefs about breast cancer, respondents were classified into two groups: Group A, including women who had not been diagnosed with breast cancer (98%, *n* = 1118, referred to as “general population”), and Group B, including women who had already been diagnosed with breast cancer (2%, *n* = 26).

Table 1 (Section 1) specifically investigates the socio-demographic characteristics of the participants. In the age distribution, a noticeable difference was observed: the majority of women in Group A were in the age brackets of 20–29 (24%) and 30–39 (38%), while the largest segments for Group B were those aged 40–49 (23%) and 50–59 (31%) (*p* < 0.001).

Group A and Group B demonstrated notable similarities across various socio-demographic characteristics. The majority of participants for both groups were from the North. When considering their educational background, both groups predominantly held a junior high school diploma. Delving into employment aspects, Group A primarily consisted of workers (28%) and students (18%). Group B showcased a diverse occupational distribution with retirees representing 27%, and workers comprising 15%. Additionally, the average duration of their stay in Italy was fairly comparable, with Group A averaging 17.95 years (SD: 9.64) and Group B at 21.34 years (SD: 14.09). However, a distinct disparity surfaced in the realm of marital status: Group A had a higher portion of married individuals (47%) relative to Group B (35%).

Table 2 (Section 2) provides insights into Chinese women’s experiences and access to health services in Italy. Both groups comprised individuals who were enrolled in the NHS. However, in terms of the difficulties encountered during registration, notable disparities emerged. Eighty-eight percent of Group A reported no difficulties, while only 62% of Group B shared the same sentiment (*p* < 0.001).

In the last two years, the majority of both groups sought the services of a general practitioner, pediatric service and ‘CUP’, which refers to ‘Centralized Booking Center’, a system used nationwide to centrally manage and book medical appointments and tests in hospitals. Among the participants, Group B, those diagnosed with breast cancer, exhibited higher utilization of healthcare services compared with Group A. Specifically, 31% of Group B had accessed emergency room services compared with just 16% from Group A (*p* = 0.04). Similarly, Group B’s engagement with the vaccination outpatient clinic was notably higher, at 69%. This is not surprising, given that individuals with pre-existing conditions, who may require more medical attention, tend to have a higher likelihood of receiving vaccinations, including the COVID-19 vaccine, especially during the COVID-19 pandemic. In contrast, only 17% of Group A used this service (*p* = 0.01). In terms of other services, 50% of Group B sought them in the last two years, as opposed to 27% in Group A (*p* = 0.008).

In their engagement with healthcare services, Group B showed a heightened frequency in visiting their general practitioner, with only 15% never visiting, which is expected as individuals with pre-existing conditions or illnesses tend to seek medical care more frequently than those in Group A (34% of whom never visited their general practitioner).Additionally, 42% of Group B had visited once, contrasting with the 22% of Group A (*p* = 0.01). Both groups expressed similar dissatisfaction levels with primary care, yet they faced varying challenges. For instance, doctor’s timings were more inconvenient for Group A (26%) than Group B (15%). However, Group B struggled more with understanding prescriptions (15%) and faced language barriers (15%), against 3% and 7%, respectively, for Group A. A majority of Group A (61%) encountered no issues, whereas this was lower for Group B at 42% (*p* = 0.006).

Regarding emergency services, 71% of Group A had never visited the emergency room, starkly differing from the mere 8% in Group B. In fact, 46% of Group B had been to the emergency room between 2 to 5 times in the past year, significantly higher than the 8% of Group A (*p* < 0.001). While 77% of Group A expressed dissatisfaction with emergency room services, this sentiment was less pronounced in Group B, at 54%, with an additional 12% from Group B showing relative satisfaction, compared with only 3% from Group A (*p* = 0.01).

With regard to gynecological services, awareness was more pronounced in Group A, with 76% being aware of the clinic’s existence versus 54% in Group B. Nevertheless, engagement was higher in Group B, with 35% having used the services in contrast with the 12% from Group A. Furthermore, Group B frequently sought pregnancy checks (19%) and gynecological visits (15%), overshadowing Group A’s figures of 4% and 5%, respectively (*p* < 0.001). While the majority of Group A (82%) were neutral about their satisfaction with the clinic, Group B displayed higher satisfaction rates, with 23% finding it quite satisfactory, as opposed to 6% in Group A (*p* < 0.001).

In this manuscript, the term ‘clinical control’ specifically refers to clinical screening examinations aimed at the early detection and monitoring of diseases. Table 3 (Section 3) presents the clinical breast cancer controls and adherence to the screening program among respondents aged 50–69, who are eligible for the free screening program. Incredibly, a majority reported having never or rarely undergone a clinical control (90%), with the adherence rate less than 10%. (Figure 1a). Distinct differences emerge when comparing Group A, representing general women aged 50–69, and Group B, which includes women of the same age but with a diagnosed tumor (*p* < 0.001). A significant 48% of respondents from Group A had never undergone a clinical exam, in contrast with a mere 8% from Group B. Regularity in clinical checkups also varied: 34% of Group A attended controls every 2 years, aligning with only 8% of Group B. Notably, annual controls were more prevalent in Group B (54%) compared with Group A (15%). Moreover, while monthly controls were rare, 1% of Group A and a significant 31% of Group B reported such frequency. (Figure 1b).

Additionally, when inquired about whether they had availed of the free screening services provided by the LHA, an impressive 96% of women over 50 have never taken advantage of these checkups (Figure 2a). Despite free screenings being offered, the vast majority of the general population does not avail of them, whereas those diagnosed with tumors seem more inclined, albeit not entirely, towards frequent screenings (*p* = 0.04). Alarmingly, only a mere 4% of Group A and 15% of Group B have opted for these checkups (Figure 2b). The lack of respondents, indicating unfamiliarity with these services in both groups, intensifies the urgency of this situation. Delving into the reasons behind the alarming low adherence to the free screening program, another layer of complexity emerges from the data on communications from the LHA. An overwhelming 72% of the general female population aged 50–69 claimed they had not received a letter from the LHA for a preventive visit. This percentage slightly drops to 46% for Group B, the subgroup of women with a previous tumor diagnosis. Only 1% confirmed being invited for breast cancer screening. Furthermore, a similar result (2%) was found for colorectal cancer screening and (even worse) the complete absence of invitations for cervical cancer screenings was declared, which is particularly concerning. For those who did receive a letter, comprehension appears to be another major roadblock. An impressive 78% of the general population expressed ambiguity or uncertainty about the content of the letter. This is not drastically different for Group B, where 38% could not fathom the letter’s intent.

Table 4 (Section 4) delves deep into the approach towards BSE among Chinese women. Notably, a significant difference emerges when comparing the two groups. A 91% of Group A and 65% of Group B indicated familiarity with BSE (*p* < 0.001). However, only 5% of Group A and 12% of Group B correctly identified that BSE is a self-examination of the breast (*p* < 0.001). The belief in its efficacy to prevent breast cancer was high in both groups (93% Group A and 77% Group B) with statistical difference. Notably, Group B thought it to be less effective than Group A (*p* = 0.002).

On the perception that BSE is unnecessary if regular mammograms are being conducted: a mixed response was seen in Group A, with 65% being uncertain, while 17% agreed with the statement. Group B had a more evenly distributed response, with 42% disagreeing and 31% agreeing (*p* < 0.001). There is also variance in opinion about the efficacy of BSE in reducing mortality rates. BSE acts as a self-assessment method that might help in identifying noticeable nodules, paving the way for timely diagnosis and action. Frequent self-checks can also aid in the early detection of a breast lump, ensuring prompt measures to mitigate the cancer’s impact. Early diagnosis is pivotal in diminishing mortality. Surprisingly, over half of the women, including those with a history of cancer (50%), concurred with the notion that early diagnosis is crucial for reducing mortality. In addition, on the effectiveness of monthly BSE in detecting lumps, only 9% of Group A and 35% of Group B agreed (*p* < 0.001).

For the frequency of BSE, the majority (65%) of women in the general group reported never practicing it, while only 4% in the group with a diagnosed tumor shared the same sentiment. In the latter group, a considerable 54% claimed to often carry out the BSE, contrasting sharply with the 5% from the general group (*p* < 0.001). Figure 3 highlights these results.

When asked about reasons for not conducting BSE, the top reason for the general group was feeling it was unnecessary (25%), followed by a belief of not being at risk (8%). Comparatively, in the group with tumors, almost half (46%) said they conduct it, revealing a significant discrepancy in awareness or commitment to this preventive measure between the two groups (*p* < 0.001).

The perception of BSE Is also notable. An overwhelming 89% of the general group agreed with the statement, “When I perform BSE, I take care of myself,” mirrored by 50% in the tumor group (*p* < 0.001). On the flip side, while the majority of the general group (63%) were uncertain as to whether BSE is embarrassing, 46% of the tumor group outright disagreed. Similarly, while many in the general group were unsure if BSE is time-consuming (64%), 42% of those with tumors disagreed, emphasizing the significant perceptual differences (both *p* < 0.001).

Regarding the capability to correctly execute BSE, 75% of the general group expressed uncertainty, whereas 50% in the tumor group shared the same sentiment. Interestingly, an impressive 91% of the general group expressed a desire for more information on BSE, significantly higher than the 68% from the tumor group (*p* < 0.001).

Lastly, when seeking guidance on BSE, the general group predominantly looked to psychologists (69%), whereas women with tumors leaned more towards breast specialists (42%). The variance in preferred information sources between the two groups is notable and statistically significant (*p* = 0.001).

The results from Section 5 of Table 4 provide an enlightening perspective on the awareness and utilization of breast prevention apps among Chinese women. A significant 91% of the general population reported being unaware of or not using these dedicated applications. Conversely, in Group B, the awareness or usage rate is slightly higher at 31% (*p* < 0.001). In essence, the results underscore a consistent trend: while there is a significant difference in overall awareness or usage (*p* < 0.001), specific app usage rates remain similarly low across both groups.

Analyzing the perceptions on breast cancer causes and symptoms between Group A and Group B, interesting insights were observed (refer to Table 5, Section 6). As expected, women from Group A appeared to be less knowledgeable compared with those who had faced the disease firsthand (91% vs. 23% cumulative for the “very” and “fairly” informed categories, respectively), (*p* < 0.001).

When probed about the origins of breast cancer, various factors were discussed, such as endocrinological causes, past mammary ailments, diet, environmental contaminants, and psychological distress. Notably, fewer women from Group B (81%) believed in the genetic connection to breast cancer compared with their Group A counterparts (94%); however, in both groups there is a widely held belief that breast cancer has a strong genetic linkage. While not every breast ailment translates to inherited cancers, and previous afflictions do not necessarily lead to cancer in the same patient, when inquired about the influence of prior breast disease on cancer risk, affirmative answers came from both cohorts (71% in Group A vs. 65% in Group B).

Although established research shows that elevated levels of sex hormones can trigger familial cancers such as breast or prostate, this correlation is denied by many women. However, a dominant part of both groups acknowledged the potential of endocrine factors in breast cancer occurrence (86% and 85% for Group A and B, respectively).

The role of nutrition in cancer risk is another point of contention (*p* < 0.001). Diet link to cancer is a recurrent media topic; however, shockingly, less than half of Group A (23%) and just 65% of Group B recognized diet significance in cancer genesis.

Environmental contaminants and pollution as cancer triggers are also debated topics. The findings show that women with cancer history more frequently acknowledge this link compared with Group A (83% vs. 54%) (*p* < 0.001).

Lastly, in regard to psychological stress, a majority of Group B (81%) see psychological stress as a pivotal cancer factor, while only half of Group A share this sentiment (*p* = 0.004). On other potential causes, both groups were fairly evenly split, without notable disparities. Roughly half of both groups believed in other unspecified causes of breast cancer, with no significant difference observed.

Inquiries on cancer symptom awareness revealed a gamut of perceptions. Questions revolved around mastodynia, palpable lumps, alterations in breast appearance, and nipple changes/discharge as potential cancer indicators. Mastodynia is not one of the symptoms found in this disease unless tissue inflammation is also present [35,36] with this occurring in only 5% of cases [11]. Women of both groups were found to be uninformed about this aspect, collecting 63% and 54% of “yes” responses for Group A and Group B respectively. In contrast with breast pain, the presence of a palpable lump may instead be an indication of breast cancer. Women with a history of previous cancer, understandably, exhibit greater awareness of this condition, with 23% of responses compared with 12%.

Few respondents (below 10%) from both groups regarded changes in breast aesthetics or nipple alterations as clear cancer signs. These changes are not directly linked to cancer and are more frequent in elderly patients.

In Section 7 of Table 5, the potential misinformation surrounding prevention is explored. Participants were questioned about their perceived level of knowledge and the perceived usefulness of common clinical screenings, such as clinical palpation, ultrasound, mammography, and other diagnostic tests like blood tests and imaging. Though the majority grasped the concept of prevention—seeing it as a combination of early diagnosis and risk factor mitigation—it is concerning that a significant portion (91%) of the general female population expressed limited or no understanding of prevention.

Common screenings like clinical palpation, mammography, and ultrasound were universally deemed useful by most participants.

As already pointed out, in Italy, mammography is indicated in women 40 years of age and older and is offered free of charge by the NHS to women over 50 and up to 69 years of age. Generally, it is not suggested for those under 30. Notably, only 51% of Group A and 58% of Group B felt that the age range of 50–69 was the appropriate period for mammography, aligning with Italy’s free screening initiative.

Magnetic resonance imaging (MRI) and computed tomography (CT) scans are typically not employed as standard screening tools, save for specific cases, such as young women with significant genetic predispositions or for specific staging purposes. However, there were stark contrasts in the opinions of Groups A and B on the utility of these tests, and the differences were statistically noteworthy (*p* = 0.009 for MRI and *p* = 0.007 for CT).

While biopsies are critical for diagnosing breast cancer, they are not standard screening procedures for preventive measures. Seventeen percent of Group A and an impressive 50% of Group B regarded these tests as beneficial for screening (*p* < 0.001).

Blood tests are also not a recognized diagnostic method for breast cancer. However, intriguingly, half of the women with a cancer diagnosis found them beneficial, compared with just 25% of the broader female population (*p* = 0.003).

Regarding oncologist consultations, an oncologist typically becomes involved post-diagnosis, not during the screening phase. However, a majority from both groups felt that an oncologist’s intervention was essential during the screening phase.

## 4. Discussion

The purpose of this study was to explore the attitudes, behaviors, and knowledge of Chinese women residing in Italy concerning breast cancer prevention, focusing particularly on screening adherence, BSE practices, the influence of technology, and misconceptions surrounding risk factors and symptoms. Understanding these dimensions is pivotal for designing effective health promotion campaigns tailored to this specific population. Respondents were divided into Group A (general population) and Group B (patients diagnosed with breast cancer) and differences were assessed.,

The socio-demographic characteristics of the participants (Table 1, Section 1) reveal several key differences and similarities between Group A and Group B. While the groups shared comparable geographical residences and educational qualifications, there were marked differences in marital and occupational statuses. Group B exhibited a noticeably higher percentage of divorced individuals (35%) compared with Group A (5%). This divergence in marital status, besides being possibly related to different age distributions in the two groups, might also be linked to the emotional, psychological, and physical toll that a cancer diagnosis and subsequent treatment can inflict on personal relationships and life dynamics. This observation aligns with findings from a population-based study, which noted that marital status significantly affects the survival rates of patients with metastatic breast cancer, in turn suggesting that the support structures inherent in marital relationships might contribute to better health outcomes. The study further highlighted the role of access to treatments such as chemotherapy and surgery, mediated by marital status, in improving survival rates, underlining the complex interplay between social support structures and health outcomes in cancer patients [37]. Concurrently, Group B’s varied occupational profile, with a significant portion being retirees (27%) and homemakers (19%), can be interpreted in the light of the group’s older age distribution. Furthermore, the challenges associated with a cancer diagnosis might push individuals towards early retirement, or to prioritize familial responsibilities and personal health over formal employment. In essence, both the direct implications of a cancer diagnosis and age-related factors might underpin the distinctions in marital and occupational patterns between the two groups.

Table 2 (Section 2) collects data on experiences and healthcare utilization in Italy. It is not surprising that those diagnosed with breast cancer would have a higher frequency of medical consultations due to the requirements of their health condition. However, a particularly striking observation is the variance in health service access between Group A and Group B. The data reveal distinct disparities in healthcare engagement between the two groups of Chinese women in Italy. Group B, those diagnosed with breast cancer, exhibited a more frequent engagement with healthcare services, particularly with general practitioners and emergency rooms. Their heightened healthcare interactions might reflect the complexities and urgencies associated with a breast cancer diagnosis. Moreover, while both groups faced challenges with healthcare services, Group B encountered more language barriers and difficulties in understanding prescriptions, emphasizing the heightened vulnerability and potential communication gaps this group experiences. In line with the literature [38], the divergent comprehension levels regarding the LHA outreach indicate the necessity for improved communication strategies tailored to the needs and comprehension levels of non-native speakers, especially for those grappling with significant health concerns like breast cancer.

The data presented in Table 3 (Section 3) offer a concerning and illuminating perspective on the patterns of breast cancer screening among Chinese women aged 50–69 in Italy, highlighting significant disparities in screening adherence between the general population and those already diagnosed with breast cancer. A distressing majority of the respondents revealed that they infrequently or never undergo breast cancer screenings. Perhaps the most alarming observation is the sheer underutilization of free screening services provided to this demographic. Even when the potential financial barrier is removed, the majority remains resistant or unaware of the screenings, underlining deeper challenges beyond cost. One plausible explanation for this trend could be the lack of effective communication from health authorities. Many women reported either not receiving an invitation letter from the LHA or, if they did, finding its content challenging to understand. This lack of clear, accessible communication could be a significant hinderance to the participation of these women in the screening programs [38].

While both groups show a disappointing uptake of the free screening services, Group B’s adherence, despite their prior diagnosis, is impressively low at just 12%. This indicates that, even among those who should be most alert to the benefits of regular screening, only a few are leveraging the freely available preventive tools. These alarming trends could be attributed to a blend of factors. Cultural perceptions, lack of adequate awareness, logistical challenges, and possible language barriers might be playing significant roles in these worrisome statistics. Reflecting our findings, other studies [39,40] emphasize the profound effect of linguistic and cultural barriers on breast and cervical cancer screening uptake, highlighting the necessity for culturally sensitive communication strategies to improve access. It is also possible that the benefits and existence of the free screening programs are not well-publicized or understood. The almost non-existent uptake of free screening services, even among those with prior diagnoses, demands urgent attention. This situation warrants immediate, robust interventions, not only to raise awareness but also to address the multifaceted barriers that deter these women from timely screenings. Health professionals, communities, and policymakers must unite in this endeavor to refocus the narrative towards prevention and early detection.

Table 4 (Section 4) highlights BSE practices, awareness, and perceptions among Chinese women in Italy. In line with other studies [41,42,43,44,45], while a significant percentage from both groups indicated familiarity with BSE, it is alarming to note the substantial disconnect between awareness and a correct understanding of what BSE entails. The high percentage of women in both groups believing in the efficacy of BSE for preventing breast cancer, contrasted with a very limited number accurately identifying BSE as a self-examination of the breast, reveals a crucial gap in knowledge dissemination. It is noteworthy that Group B showed lesser belief in the efficacy of BSE than Group A. This could be a reflection of their lack of awareness regarding BSE or, simply, it could be attributed to their lived experiences, where perhaps BSE did not play a pivotal role in their initial diagnosis, or they might be more informed about the limitations of BSE post-diagnosis. The mixed views on the necessity of BSE when regular mammograms are being conducted further highlight the ambiguous perceptions surrounding this preventive measure. The significantly higher frequency of BSE among women with tumors may reflect a reactionary behavior post-diagnosis, emphasizing the need for proactive healthcare practices among the general population. The reasons cited for not conducting BSE, especially the feeling that it is unnecessary or that the subjects are not at risk, indicate a potential underestimation of breast cancer risks among the general female population, as highlighted in literature [46].

Emotional resonance on BSE was also assessed. The perception that BSE is a form of self-care was overwhelmingly shared by the general group, suggesting that BSE is not just seen as a clinical act but also carries emotional and psychological significance. The contrasting views on whether BSE is embarrassing or time-consuming between the two groups could be influenced by their different experiences and knowledge levels.

The need expressed by a vast majority for more information on BSE highlights the existing information gap. Moreover, the preference of women from the general group to seek guidance on BSE from psychologists rather than medical professionals is intriguing. In contrast, women with tumors, having navigated the medical system, seem to recognize and prioritize the expertise of breast specialists. Efforts should focus on not just promoting the practice of BSE, but also ensuring that it is understood and performed correctly. Other studies in literature have similarly highlighted the emotional and psychological dimensions of BSE. For instance, research to assess predictors of BSE behavior among female university students have revealed that perceived benefits, severity, and self-efficacy significantly influence BSE engagement. This underscores the role of individual belief in one’s ability to perform BSE, pointing to the deeper psychological underpinnings of this health practice [47]. Another study examining BSE knowledge and practices among female students highlighted barriers to BSE practice, such as a lack of correct procedural knowledge and the perception of BSE as unnecessary without existing breast problems, further emphasizing the need for education that bridges emotional support and technical know-how [48]. These insights complement the observation that the perception of BSE as a form of self-care was overwhelmingly shared by the general group, suggesting that BSE is not just seen as a clinical act but also carries emotional and psychological significance.

Table 4 (Section 5) collects awareness and utilization of breast prevention apps among Chinese women. The digital age has brought forth numerous technological tools designed to promote health awareness and proactive healthcare behaviors. Among these tools, health applications, especially those dedicated to breast cancer prevention, have emerged as potential aids in early detection and education [49]. Our findings highlight a notable disparity in awareness of breast prevention apps between women from the general population and those diagnosed with tumors. The heightened awareness among the latter could be attributed to their more active engagement with breast health post-diagnosis, potentially leading them to explore various resources. However, a striking observation emerges when one delves into the utilization rates. Despite the differential in awareness, the actual usage of these apps remains consistently low across both groups. Research supports the observation that while there is a growing awareness of breast cancer prevention apps among women, actual usage rates remain low [50]. To bridge the gap between awareness and active use, there is a pressing need for a multifaceted approach that combines technological innovation with cultural competence, user education, and trust-building measures.

The perceptions surrounding the causes and symptoms of breast cancer, as revealed by the results shown in Section 6 of Table 5, offer valuable insights into the understanding and beliefs of Chinese women in Italy. These perceptions, which vary significantly between the general population and those diagnosed with breast cancer, carry implications for awareness campaigns, patient education, and healthcare delivery. The relatively lower belief in genetic causation of breast cancer among Group B compared with Group A is intriguing. This might reflect a more nuanced understanding among those diagnosed with breast cancer, recognizing that, while genetics plays a role, it is not the sole factor. Research on the perceptions surrounding the causes and symptoms of breast cancer aligns with these observations. Women with breast cancer are more likely to attribute their condition to mental or emotional factors, such as stress, while control women more commonly cite familial or inherited factors [51]. At the same time, the affirmative responses from both groups about the influence of prior breast disease on cancer risk suggest a common belief in cumulative breast health. While not entirely accurate, this belief might stem from a general understanding that any ailment or irregularity in an organ could predispose it to further issues.

The broad acknowledgment of endocrine factors in breast cancer occurrence is an encouraging sign, indicating that certain scientific understandings have successfully reached the public. However, the varied recognition of diet’s role in cancer genesis highlights the need for enhanced clarity in public health messaging, especially given the modifiable nature of dietary habits [28,52]. While diet, as a modulable factor for cancer prevention, is often the focus of media attention, the relationship between environmental pollution and the occurrence of breast cancer is currently highly debated in the scientific community [30,53]. Group B’s heightened recognition of environmental contaminants suggests that firsthand experience with the disease may lead to a more encompassing view of potential causal factors. Similarly, psychological stress is also considered a possible cause of cancer [26,54,55] and their emphasis on this factor risk, possibly stems from the personal challenges and realizations encountered during their diagnosis and treatment phases. In terms of symptom awareness, prevalent misconceptions exist against the literature [41,56,57,58,59], particularly regarding the association of mastodynia with breast cancer. However, the more accurate recognition of palpable lumps by women with a breast cancer history underscores the transformative impact of personal experience on health awareness. Collectively, these findings spotlight the imperative to bridge knowledge gaps while capitalizing on accurate public understandings to foster proactive breast health behaviors.

Section 7 of Table 5 reveals concerning misconceptions surrounding breast cancer prevention among Chinese women. A majority recognize the value of screenings like mammography and clinical palpation. However, some troubling disparities emerge between the general female population and those diagnosed with breast cancer. Notably, both groups exhibit misunderstandings about the recommended age range for mammography in Italy. Additionally, perceptions about specialized tools like MRI and CT scans differ significantly between the groups, even though these are not standard screening tools. The overconfidence in biopsies and blood tests, especially among diagnosed women, further highlights potential misinformation. Other studies in the literature have also underscored the divergence between public perceptions and scientifically recognized risk factors for breast cancer [51].

Most intriguingly, both groups express a desire for oncologist involvement during the initial screening phase, a clear departure from standard medical practice.

In comparing the findings of our current study with those presented in [32], which surveyed the broader Italian population, notable differences emerge, particularly in the area of health screening participation. Our analysis reveals that the rate of participation in health screenings among Chinese women residing in Italy is significantly lower than that observed in the general Italian population. This discrepancy highlights a critical gap in health promotion and education efforts targeting this specific demographic.

In essence, while there is an overall appreciation for preventive measures, there is a pressing need to address and rectify these misconceptions, especially considering the contrasting views between the two groups.

## 5. Limits

The results of the study must be considered while taking into account some limitations. Firstly, there might have been cultural or linguistic nuances that, despite the questionnaire being translated by a native speaker, could have been overlooked or misinterpreted. This might have influenced the way respondents understood or answered certain questions. Another limitation is the potential non-representation of a broader Chinese demographic, given that those who are less integrated or less proficient in the dominant language might not have participated. This limitation is surely related to the mode of administration through the telematic medium, which is probably utilized more by younger women. Another limitation mainly concerns the choice of electronic dissemination of the questionnaire that may have partially excluded those who had little computer background. Additionally, there might have been some respondents who, due to cultural or personal reasons, preferred not to participate in such surveys, leading to potential non-response bias. Possible information bias may be due to a reluctant attitude to declare and therefore admit a lack of knowledge of the phenomenon.

## 6. Conclusions

Our research underscores a pronounced gap in breast cancer screening adherence and knowledge among Chinese women residing in Italy. Particularly evident is the discrepancy between the general population and those who have previously encountered cancer. The pressing necessity to enrich these women with comprehensive information to facilitate better screening participation is evident. Despite the global strides in breast cancer awareness and management, the suboptimal participation in screenings remains a concern. Factors ranging from limited public awareness to socio-psychological barriers might be influencing these numbers. Recognizing and mitigating cancer risk factors is a potent tool, especially for asymptomatic women. Hence, promoting awareness and participation in screening practices, especially mammography, for age-appropriate women is paramount.

When delving into self-examination practices, a striking observation emerges: a majority of women do not know what BSE is, either do not practice it, or lack the correct methodology. This paradox emphasizes the need for effective educational programs. Given that BSE can commence as early as 20 years of age, there is an opportunity to inculcate this practice among younger women, potentially through educational initiatives in academic institutions.

Embracing technology could further bolster these efforts. Furthermore, nurses could play a pivotal role in these educational endeavors, enhancing their stature in preventive healthcare. The study also highlights pervasive knowledge deficits across various domains, from symptom recognition to understanding causative factors and preventive measures. To navigate these challenges, the leveraging of specialized centers dedicated to early breast cancer diagnosis and treatment, such as breast units [60] in the territory, could be transformative. With over 200 breast units established, as highlighted by the state–regions conference, these units not only cater to diagnosed patients, but also play a pivotal role in preventive education for asymptomatic women, promoting healthy lifestyles and facilitating access to diagnostic tests when needed. In conclusion, it is evident that there is an exigent need to elevate health literacy among Chinese women in Italy concerning breast cancer risk factors, symptoms, and prevention. As breast cancer continues to be a leading cause of distress and mortality among women globally, such targeted, culturally sensitive interventions are not just desirable but imperative.

## Figures and Tables

**Figure 1 ijerph-21-00308-f001:**
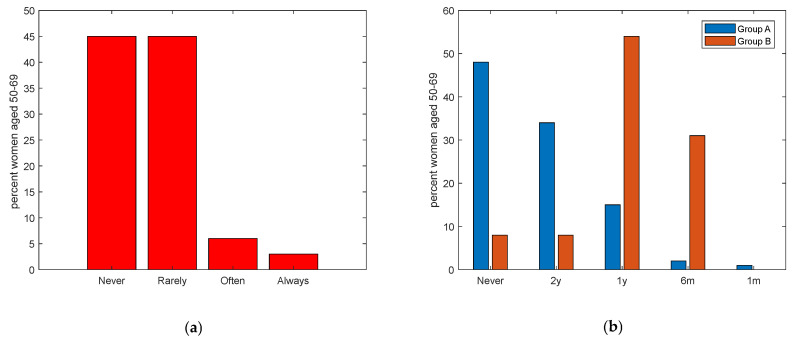
(**a**). Frequencies of clinical controls for breast cancer for women aged 50–69 years (*p* < 0.001). (**b**). Frequencies of clinical controls for breast cancer for women aged 50–69 years divided into Group A (women of the general population) in blue, and Group B (women already diagnosed with breast cancer) in red. (y = years, m = months).

**Figure 2 ijerph-21-00308-f002:**
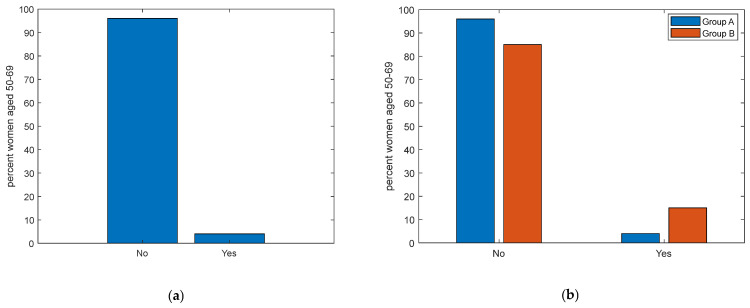
(**a**). Frequencies of adherence to free screening programs for women aged 50–69 years (*p* < 0.001). (**b**). Frequencies of adherence to free screening programs for women aged 50–69 years, divided into Group A (women of the general population) in blue, and Group B (women already diagnosed with breast cancer) in red.

**Figure 3 ijerph-21-00308-f003:**
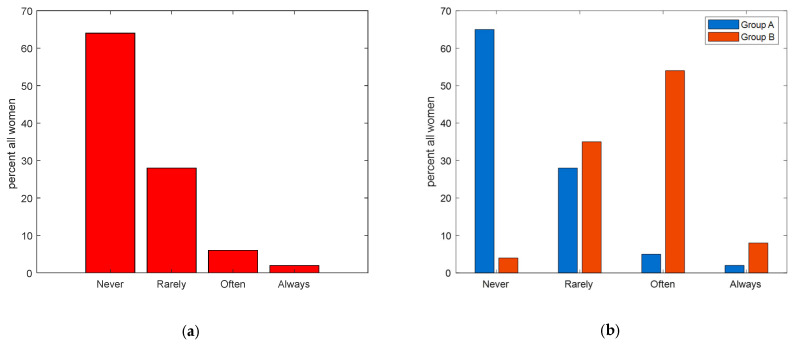
(**a**) Frequencies of BSE practices among all respondents. (**b**) Frequencies of BSE practices divided into Group A (women from the general population) in blue, and Group B (women already diagnosed with breast cancer) in red.

**Table 1 ijerph-21-00308-t001:** Socio-demographic characteristics of the study participants. It compares the general population of Chinese women in Italy (Group A) with those who have been diagnosed with breast cancer (Group B). Differences between the groups are indicated with *p*-values, where a value less than 0.05 indicates statistical significance; *** *p* < 0.001).

Section 1: Socio-Demographic Characteristics	Group A: Individuals from the General Population (98%, *n* = 1118)	Group B: Patients with Breast Cancer (2%, *n* = 26)	*p*-Value
Age			
20–29	263 (24)	2 (8)	<0.001 ***
30–39	427 (38)	5 (19)	
40–49	211 (19)	6 (23)	
50–59	160 (14)	8 (31)	
60–69	57 (5)	5 (19)	
Geographical Area			0.81
North	377 (34)	18 (69)	
Center	422 (38)	6 (23)	
South and Islands	319 (29)	2 (8)	
Marital Status			0.53
Married	528 (47)	9 (35)	
Divorced	54 (5)	9 (35)	
Single	446 (40)	3 (12)	
Separated	73 (7)	1 (4)	
Widowed	17 (2)	4 (15)	
Educational level			0.85
Degree	36 (3)	3 (12)	
High school graduation	184 (16)	3 (12)	
Junior High School Diploma	696 (62)	14 (54)	
Elementary Education	160 (14)	3 (12)	
None	42 (4)	3 (12)	
Occupational Status			0.56
Worker	310 (28)	4 (15)	
Housewife	194 (17)	5 (19)	
Public Employee	62 (6)	1 (4)	
Freelancer	92 (8)	2 (8)	
Student	197 (18)	2 (8)	
Retired	113 (10)	7 (27)	
Other	150 (13)	5 (19)	
Unemployed	0	0	
Are you currently working?			0.94
No	466 (42)	11 (42)	
Yes	652 (58)	15 (58)	
Years in Italy			0.298
Range	1–50	2–44	
Mean	17.95	21.34	
SD	9.64	14.09	

**Table 2 ijerph-21-00308-t002:** Respondents’ access to health services, divided into Group A (women in the general population, *n* = 1118) and Group B (women with cancer, *n* = 26). Differences in responses were evaluated. A *p* value < 0.05 is considered statistically significant (* *p* < 0.05; ** *p* < 0.01; *** *p* < 0.001).

Section 2. Access to Health Services	Group A: Individuals from the General Population (98%, *n* = 1118)	Group B: Patients with Breast Cancer (2%, *n* = 26)	*p*-Value
Are you enrolled in the National Health Service (NHS)?			0.83
No	2 (0)	0	
Yes	1116 (100)	26 (100)	
Did you encounter any difficulties in enrollment?			<0.001 ***
I do not have a residence permit	43 (4)	5 (19)	
I tried but had difficulty	17 (2)	1 (4)	
I do not know how to do it	33 (3)	2 (8)	
I never thought about it	20 (2)	1 (4)	
I do not care	16 (1)	1 (4)	
I had no difficulty	989 (88)	16 (62)	
During the past two years, have you relied on the services of your primary care physician?			0.27
No	243 (22)	8 (31)	
Yes	875 (78)	18 (69)	
Over the past two years, have you relied on the services of a pediatrician?			0.41
No	850 (76)	8 (31)	
Yes	267 (24)	18 (69)	
Over the past two years, have you relied on the services of the emergency room (ER)?			0.04 *
No	941 (84)	18 (69)	
Yes	177 (16)	8 (31)	
Over the past two years, have you relied on the services of a hospital?			0.07
No	923 (83)	18 (69)	
Yes	195 (17)	8 (31)	
Over the past two years, have you relied on the services of a gynecological consultatory?			0.70
No	853 (76)	19 (73)	
Yes	265 (24)	7 (27)	
Over the past two years, have you relied on the services of the centralized booking center (CUP) service?			0.45
No	923 (83)	20 (77)	
Yes	195 (17)	6 (23)	
Over the past two years, have you relied on the services of the vaccine outpatient clinic?			0.01 *
No	923 (83)	8 (31)	
Yes	195 (17)	18 (69)	
Over the past two years, have you relied on anything else?			0.008 **
No	819 (73)	13 (50)	
Yes	299 (27)	13 (50)	
Have you chosen your primary care physician?			<0.001 ***
No	74 (7)	7 (27)	
Yes	1044 (93)	19 (73)	
In the past year, how many times have you relied on your family physician?			0.01 *
Never	384 (34)	4 (15)	
1 time	241 (22)	11 (42)	
2–5 times	399 (36)	9 (35)	
>5 times	94 (8)	2 (8)	
Are you comfortable with your family physician?			0.28
Not at all	40 (4)	1 (4)	
Little	628 (56)	12 (46)	
I do not know	64 (6)	3 (12)	
Quite	343 (31)	8 (31)	
Very	43 (4)	2 (8)	
What problems are there to report about the family physician?			0.006 **
Schedules do not fit	291 (26)	4 (15)	
I have difficulty understanding the recipes	36 (3)	4 (15)	
We do not understand each other because of the language	83 (7)	4 (15)	
I have never had any problems	29 (3)	3 (12)	
More	679 (61)	11 (42)	
In the past year, have you relied on the services of the CUP service?			0.08
No	853 (76)	16 (62)	
Yes	265 (24)	10 (38)	
What problems are there to report for CUP?			0.005 **
It is not clear how it works	83 (7)	8 (31)	
It was difficult to book	138 (12)	3 (12)	
I have never had any problems	74 (7)	3 (12)	
More	823 (74)	12 (46)	
In the past year, how many times have you relied on the emergency room?			<0.001 ***
Never	793 (71)	2 (8)	
1 time	229 (20)	12 (46)	
2–5 times	92 (8)	12 (46)	
>5 times	4 (0)	0	
Were you satisfied with the service?			0.01 *
Not at all	39 (3)	3 (12)	
Little	866 (77)	14 (54)	
I do not know.	114 (10)	2 (8)	
Quite	65 (6)	4 (15)	
A lot	34 (3)	3 (12)	
What problems are there to report about the emergency room?			<0.001 ***
The operators did not have time to explain	68 (6)	6 (23)	
We did not understand each other because of the language	87 (8)	5 (19)	
It is unclear how access works	51 (5)	2 (8)	
Other	912 (82)	13 (50)	
Do you know that there is a night and holiday medical service?			<0.001 ***
No	105 (9)	8 (31)	
Yes	1013 (91)	18 (69)	
In case of need, would you know how to contact this medical service?			0.73
No	335 (30)	7 (27)	
Yes	783 (70)	19 (73)	
Are you aware of the existence of the counseling center?			<0.001 ***
No	845 (76)	12 (46)	
Yes	273 (24)	14 (54)	
Have you ever used the counseling center?			<0.001 ***
No	985 (88)	17 (65)	
Yes	133 (12)	9 (35)	
If yes, for what reason?			<0.001 ***
Psychological assistance	17 (2)	1 (4)	
PAP test	8 (1)	0	
Contraception	33 (3)	2 (8)	
Scheduled checks in pregnancy	47 (4)	5 (19)	
Termination of pregnancy	13 (1)	2 (8)	
Gynecological examination	59 (5)	4 (15)	
Other	941 (84)	12 (46)	
Were you satisfied with the service of the counseling center?			<0.001 ***
Not at all	31 (3)	3 (12)	
Little	75 (7)	1 (4)	
I do not know.	914 (82)	13 (50)	
Quite	32 (6)	6 (23)	
A lot	36 (3)	3 (12)	

**Table 3 ijerph-21-00308-t003:** Section 3 of the questionnaire concerns clinical breast cancer controls and adherence to screening programs offered by the NHS in adult women aged 50–69 years. Women were divided into Group A (women in the general population, *n* = 217) and Group B (women with cancer, *n* = 26). Differences in responses were evaluated. A *p* value <0.05 is considered statistically significant (* *p* < 0.05; ** *p* < 0.01; *** *p* < 0.001).

Section 3: Clinical Breast Cancer Controls and Adherence to Screening Programs, Women Aged 50–69	Group A Women in the General Population Aged 50–69 (*n* = 217) N (%)	Group B Women with Cancer Aged 50–69 (*n* = 13) N (%)	*p*-Value
Have you ever undertaken clinical controls for the early detection of breast cancer?			<0.001 ***
Never	103 (47)	1 (8)	
Rarely	102 (47)	2 (15)	
Occasionally	0	0	
Often	7 (3)	7 (54)	
Always	5 (2)	2 (23)	
If yes, please indicate the frequency			0.47
I have never had a screening exam	104 (48)	1 (8)	
Every 2 years	74 (34)	1 (8)	
Every year	32 (15)	7 (54)	
Every 6 months	4 (2)	4 (31)	
Every month	3 (1)	0	
Have you ever taken advantage of the free screening offered by the region?			0.04 *
No	209 (96)	11 (85)	
Yes	8 (4)	2 (15)	
I do not know of them	0	0	
Have you ever been called by the local health authority (LHA) for a visit dedicated to prevention?			0.73
Yes, breast cancer (mammography)	3 (1)	1 (8)	
Yes, for colorectal cancer (stool analysis)	4 (2)	1 (8)	
Yes, for cervical cancer (PAP test)	0	0	
No, I did not receive the letter	157 (72)	6 (46)	
I do not know	53 (24)	5 (38)	
If you received the letter, how understandable was it?			0.50
Not at all	8 (4)	1 (8)	
Little	40 (18)	1 (8)	
I do not know	151 (70)	7 (54)	
Quite	0	0	
A lot	18 (8)	4 (31)	
Have you ever had a biopsy?			0.47
No	154 (71)	8 (62)	
Yes	63 (29)	5 (38)	
Missing	0	0	
Have you ever had a mammogram?			<0.01 **
No	154 (71)	1 (8)	
Yes	63 (29)	12 (92)	
Missing	0	0	
Have you ever had an ultrasound?			<0.001 ***
No	191 (88)	1 (8)	
Yes	26 (12)	12 (92)	
Missing	0	0	
Have you ever had magnetic resonance imaging (MRI)?			<0.001 ***
No	201 (93)	8 (62)	
Yes	16 (7)	5 (38)	
Missing	0	0	

**Table 4 ijerph-21-00308-t004:** Section 4 (approach to BSE), and Section 5 (knowledge and use of apps dedicated to prevention) for all respondents were assessed. Women were divided into Group A (general population women, *n* = 1118) and Group B (women with tumors, *n* = 26). Differences in responses were evaluated. A *p*-value < 0.05 is considered statistically significant ** *p* < 0.01; *** *p* < 0.001).

Section 4: Approach to Breast Self-Examination (BSE)	Group A Women in The General Population (*n* = 1118) N (%)	Group B Women with Cancer (*n* = 26) N (%)	*p*-Value
Have you ever heard of BSE?			<0.001 ***
No	98 (9)	9 (35)	
Yes	1020 (91)	17 (65)	
In your opinion, what does self-examination consist of?			<0.001 ***
Breast self-examination	51 (5)	3 (12)	
Clinical examination of the breast (search for visible and/or palpable findings in the breast and surrounding areas, e.g., areas of lymphatic drainage axilla, neck)	284 (25)	7 (27)	
Radiological examination of the breast (mammography, ultrasonography, MRI, biopsy, chest X-ray, scintigraphy, CT scan, PET/CT, chest X-ray)	717 (64)	9 (35)	
I do not know	50 (4)	5 (19)	
Other	16 (1)	2 (8)	
Does self-examination help prevent breast cancer?			0.002 **
No	80 (7)	6 (23)	
Yes	1038 (93)	20 (77)	
Is self-palpation not necessary if I perform periodic mammography?			<0.001 ***
Strongly agree	52 (5)	4 (15)	
Agreed	194 (17)	8 (31)	
In disagreement	144 (13)	11 (42)	
Strongly disagree	5 (0)	0	
Uncertain	723 (65)	3 (12)	
Performing a self-examination reduces mortality.			0.18
Strongly agree	78 (7)	11 (42)	
Agree	590 (53)	13 (50)	
In disagreement	6 (1)	0	
Strongly disagree	1 (0)	0	
Uncertain	443 (40)	2 (8)	
Performing self-examination every month helps me find the nodules			<0.001 ***
Strongly agree	34 (3)	4 (15)	
Agree	104 (9)	9 (35)	
In disagreement	105 (9)	2 (8)	
Strongly disagree	7 (1)	0	
Uncertain	868 (78)	11 (42)	
How often do you perform self-palpation?			<0.001 ***
Never	731 (65)	1 (4)	
Rarely	313 (28)	9 (35)	
Occasionally	0	0	
Often	57 (5)	14 (54)	
Always	17 (2)	2 (8)	
If not, state the reason			<0.001 ***
I perform it	278 (25)	12 (46)	
I am not at risk	88 (8)	3 (12)	
I do not remember to run it	62 (6)	4 (15)	
Fear of ominous prognosis	101 (9)	2 (8)	
I do not know how to execute it properly	37 (3)	4 (15)	
I do not know what it is	32 (3)	0	
missing	0	1 (4)	
When I do self-examination, I take care of myself.			<0.001 ***
Strongly agree	75 (7)	11 (42)	
Agree	997 (89)	13 (50)	
In disagreement	32 (3)	1 (4)	
Strongly disagree	11 (1)	1 (4)	
Uncertain	3 (0)	0	
Self-palpation is embarrassing			<0.001 ***
Strongly agree	38 (3)	5 (19)	
Agree	103 (9)	8 (31)	
In disagreement	704 (63)	1 (4)	
Strongly disagree	267 (24)	12 (46)	
Uncertain	6 (1)	0	
Self-examination takes too much time			<0.001 ***
Strongly agree	52 (5)	6 (23)	
Agree	120 (11)	8 (31)	
In disagreement	715 (64)	1 (4)	
Strongly disagree	230 (21)	11 (42)	
Uncertain	1 (0)	0	
I have more important problems than self-examination			<0.001 ***
Strongly agree	48 (4)	4 (15)	
Agree	90 (8)	9 (35)	
In disagreement	877 (78)	7 (27)	
Strongly disagree	101 (9)	6 (23)	
Uncertain	2 (0)	0	
I am able to perform self-examination correctly			<0.001 ***
Strongly agree	35 (3)	3 (12)	
Agree	81 (7)	8 (31)	
In disagreement	843 (75)	13 (50)	
Strongly disagree	147 (13)	2 (8)	
Uncertain	12 (1)	0	
I would like more information about self-examination.			<0.001 ***
Yes	1013 (91)	18 (68)	
No	105 (9)	8 (31)	
Which figure do you find helpful in obtaining information about self-examination?			0.001 ***
Primary care physician	27 (2)	1 (4)	
Nurse	17 (2)	9 (35)	
Oncologist Psychologist	271 (24)	1 (4)	
Breast specialist	777 (69)	11 (42)	
Other	14 (1)	3 (12)	
Section 5: Knowledge and Use of Apps Dedicated to Prevention			
Do you know or use dedicated applications for self-examination?			<0.001 ***
No	1016 (91)	18 (69)	
Yes	102 (9)	8 (31)	
Do you use BreastTest?			0.64
No	1091 (98)	25 (96)	
Yes	27 (2)	1 (4)	
Do you use Igyno?			0.45
No	1094 (98)	26 (100)	
Yes	24 (2)	0	
Do you use Breast Cancer Indicators?			0.41
No	1090 (97)	26 (100)	
Yes	28 (3)	0	
Do you use other apps?			0.66
No	1090 (97)	25 (96)	
Yes	28 (3)	1 (4)	

**Table 5 ijerph-21-00308-t005:** Knowledge and beliefs about cancer causes and symptoms (Section 1) and knowledge and beliefs about cancer prevention (Section 2) were assessed. Women were divided into Group A (general population women, *n* = 1118) and Group B (women with tumors, *n* = 26). Differences in responses were evaluated. A *p*-value < 0.05 is considered statistically significant (* *p* < 0.05; ** *p* < 0.01; *** *p* < 0.001).

Section 6: Knowledge and Beliefs about the Causes and Symptomatology of Breast Cancer	Group A Women in the General Population (*n* = 1118) N (%)	Group B Women with Cancer (*n* = 26) N (%)	*p*-Value
Do you think you are well informed about breast cancer?			<0.001 ***
A lot	26 (2)	5 (19)	
Little	80 (7)	13 (50)	
Quite	939 (84)	6 (23)	
Not at all	73 (7)	2 (8)	
Do you think the cause of breast cancer is genetics?			0.009 **
No	71(6)	5 (19)	
Yes	1047 (94)	21 (81)	
Do you think the cause of breast cancer is endocrine?			0.82
No	155 (14)	4 (15)	
Yes	963 (86)	22 (85)	
Do you think a cause of breast cancer may be previous breast disease?			0.50
No	320 (29)	9 (35)	
Yes	798 (71)	17 (65)	
Do you think one cause of breast cancer may be radiation?			0.02 *
No	951 (85)	18 (69)	
Yes	167 (15)	8 (31)	
Do you think one cause of breast cancer may be nutrition?			<0.001 ***
No	856 (77)	9 (35)	
Yes	262 (23)	17 (65)	
Do you think a cause of breast cancer may be environmental factors and pollution?			<0.001 ***
No	929 (83)	14 (54)	
Yes	189 (17)	12 (46)	
Do you think a cause of breast cancer may be psychological stress?			0.004 **
No	532 (48)	5 (19)	
Yes	586 (52)	21 (81)	
Do you think the cause of breast cancer may be another one?			0.19
No	573 (51)	10 (38)	
Yes	545 (49)	16 (62)	
What do you think the symptoms of cancer might be?			0.44
Palpable nodule	135 (12)	6 (23)	
Change in breast shape and size	700 (63)	14 (54)	
Nipple secretion	69 (6)	2 (8)	
Nipple alteration	70 (6)	3 (12)	
Other	4 (0)	0	
I don’t know	30 (3)	1 (4)	
Section 7: Knowledge and beliefs about breast cancer prevention			
Do you feel that you are well informed about breast cancer prevention?			<0.001 ***
A lot	22 (2)	7 (27)	
Quite	69 (6)	13 (50)	
Little	955 (85)	4 (15)	
Not at all	72 (6)	2 (8)	
What does prevention mean to you?			<0.001 ***
Early detection of cancers	301 (27)	11 (42)	
Prevention of risk factors	593 (53)	5 (19)	
Prevention of complications	159 (14)	2 (8)	
Don’t know	41 (4)	6 (23)	
More	24 (2)	2 (8)	
At what age do you think mammography is recommended?			0.26
<20 years old	6 (1)	0	
20–30	41 (4)	0	
30–40	118 (11)	2 (8)	
40–50	199 (18)	4 (15)	
50–60	569 (51)	15 (58)	
60–70	165 (15)	3 (12)	
I don’t know	20 (2)	1 (8)	
How often do you think mammography is recommended?			0.0057
Based on age/familiarity	0	0	
More than every year	0	0	
Every month	11 (1)	0	
Every 6 months	66 (6)	2 (8)	
Every year	846 (76)	13 (50)	
Every 2 years	141 (13)	5 (19)	
I don’t know	54 (5)	6 (23)	
Do you consider clinical palpation useful as an act of breast cancer prevention?			0.42
No	117 (10)	4 (15)	
Yes	1001 (90)	22 (85)	
Do you think Magnetic Resonance Imaging (MRI) is useful for breast cancer prevention?			0.009 **
No	501 (45)	5 (19)	
Yes	617 (55)	21 (81)	
Do you consider biopsy useful as an act of breast cancer prevention?			<0.001 ***
No	928 (83)	13 (50)	
Yes	190 (17)	13 (50)	
Do you think computed tomography (CT) is useful for breast cancer prevention?			0.007 **
No	639 (57)	8 (31)	
Yes	479 (43)	18 (69)	
Do you think blood tests are useful for breast cancer prevention?			0.003 **
No	839 (75)	13 (50)	
Yes	279 (25)	13 (50)	
Do you find the interview with the oncologist useful as an act of breast cancer prevention?			0.13
No	84 (8)	4 (15)	
Yes	1034 (92)	22 (85)	

## Data Availability

The data presented in this study are available on request from the corresponding author.

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
