# Peer review of "Exploring the Landscape of Breast Cancer Prevention among Chinese Residents in Italy: An In-Depth Analysis of Screening Adherence, Breast Self-Examination (BSE) Practices, the Role of Technological Tools, and Misconceptions Surrounding Risk Factors and Symptoms"

_ijerph, 2024, doi:10.3390/ijerph21030308_

Round 1

Reviewer 1 Report

Comments and Suggestions for Authors

The manuscript describes a questionnaire-based survey of attitudes and behaviour of women of Chinese descent living in Italy in regards to breast screening and breast cancer knowledge.    The manuscript is generally well organized and written and is accessible to the reader.

I have only a few small concerns regarding the manuscript in its current form.  

1.  Generally, I found the introduction to be a bit long with a few parts being unnecessary for the narrative.  For example, lines 78-89 on risk (genetic, lifestyle, etc) perhaps provides more detail than is necessary to introduce the concepts being evaluated in Section 6 of the survey.  Lines 108-112 are largely reiterated in lines 113-117 (and already introduced back on lines 58-61).      I think shortening the the introduction by removing the repetition and make the details more succinct will make this a much more impactful manuscript.

2.  "snowball method" :  please define.  I'm assuming this means that the survey is passed on by the participants... is this correct?

3.   Depending on the jurisdiction, recommendations for BSE are quite varied: from "not recommended" to "strong encouragement".  The authors obviously believe that BSE is important in Italy.  However, is this an official NHS recommendation? If so, does the NHS provide educational training, etc for BSE?   Also, if there is a recommendation, is the recommendation for all women or only for specific sub groups (e.g underserved or remote areas)? 

4.  "Clinical controls":   I find this word usage a bit strange.  Generally "clinical control" has these common meanings:

i.  the (usually healthy or non-intervention) match in a "case-control" clinical trial

ii.  the measures in a clinical trial to ensure proper procedures are followed

iii.  the clinical management of disease progression (e.g. controlling epilepsy, chronic disease etc)

Here,  clinical control appears to mean something like the 3rd option, but I've never heard of it used in describing screening/early detection.    A search of google scholar of "breast clinical control" turned up only a single reference (Marchina et al Oncology 2010) that appears to use it in a similar fashion to this current manuscript.   

Because of the ambiguity,  it would be worthwhile when introducing Section 3 of the questionnaire to define or give examples of what is meant by a "clinical control" . Perhaps:   "Here, clinical control is a screening examination..."  or "Here, clinical control refers to one or more of the following...."

5.   Material and Methods:   There are similarities to some of the previous work (ref 28 Conte et al) on the broader Italian population.   It would help the reader to  show the natural progression of this work from the preceding work.  For example,- "The methodology is similar to the approach described in [ref 28] ".   Highlight the similarities/differences.   For example:  "In this study, we used the [same/different questions/expanded sections]".   Highlight similarities/differences in dissemination.

6.  I'm somewhat surprised that some comparison are not made to ref 28 in terms of results.  I realize of course that there won't be entry-by-entry correlations, but some of the broader findings would be informative.  For example in the discussion, one could show that the lack of participation in screening is much(!!) lower than the general population, highlighting the need for promotion/education/etc.   This need not be extensive,  one or two comparisons may be sufficient.

7.   The results and discussion are a bit repetitive -- largely because conclusions/discussion is interspersed throughout the results section.  For example,  Line 361:  "This suggests a broader need to promote and integrate technological tools for breast cancer prevention and awareness."  is really a discussion (or even conclusion section) and should be moved there (which is already highlighted in the discussion around line 541).

8.  p15 - the table got a little mangled in my copy.   Misordered words:  "A lot Quite" on the first part of Section 7.  Overlapping text on the section beginning with "How often" other alignment/word order issues.

Comments on the Quality of English Language

English is excellent

Reviewer 2 Report

Comments and Suggestions for Authors

I would like to thank the authors for this interesting article. They explore the breast cancer prevention, e.g., screening adherence, breast self-examination, technological tools, and misconceptions in Chinese women living in Italy. The manuscript is well written.

Please find attached some suggestions for improvement:

1)     NHS (abbreviation) is not explained in the abstract. Please add this information.

2)     Introduction: page 2, line 53: How old are the Chinese women with new diagnosed breast cancer? Is there a difference compared to Europe or the US?

3)     Introduction: page 2, line 55-57: How high is the incidence of breast cancer and mortality rate in Italy compared to China? (in percentage). Is there a difference?

4)     Introduction: page 2, line 62-66: Please add references.

5)     In general, I think that the introduction is a little bit long and might need some shortening. Furthermore, I don’t understand, why this project focuses only on Chinese women living in Italy. Why did the authors not include other immigrants, as they might have cultural, linguistic, or socioeconomic barriers as well? It would be great if this could be explained in more detail in the introduction. 

6)     Materials and Methods: How did the authors make sure that “all” the Chines women living in Italy have received the questionnaire?

7)     Materials and Methods: Why were patients older than 20 years included in the survey? While “routine” breast cancer screening starts at the age of 50?

8)     Did the authors submit the questionnaire to an institutional review board for ethics approval? The number of approvals should be added. The ethics statement at the end of the text (after conclusion) is missing, too.

9)     Is the questionnaire validated or previously used?

10)  In my opinion, it is very difficult to compare a group of 26 patients to 1118 patients. It is not surprising, that the few patients (26) needed to see a physician more often, as they had breast cancer diagnosis.

11)  The results section should just illustrate the results. But the results are already discussed and evaluated, e.g.: “This suggests potential gaps in recall, confusion, or even possible mishandling or overlooking of these communications”

12)  Table SECTION 7: In the field “How often do you think mammography is recommended”- please edit in correct form “based on age/familiarity”.

All in all, it is an interesting, well-written article, yet it requires thorough revision and condensing. In the introduction it must be explicitly explained why this questionnaire focused solely on this particular demographic. The Results Section redundantly reiterates table data in the text, resulting in unnecessary repetition. This redundancy persists into the discussion, as nearly two third of the discussion is a repetition of the results from the questionnaire. In my opinion, the discussion needs major revision. Hardly any literature is cited. However, the discussion should primarily aim to contextualize the research findings within the existing scientific discourse and contemporary studies on the subject.

Round 2

Reviewer 2 Report

Comments and Suggestions for Authors

Dear authors,

Thank you for the revision of the manuscript, which significantly improved the paper. Nevertheless, I have some concerns, that I would like to share with you.

1)     I think there might be some mistakes with the references. You added the following sentence: 

“In this study, we employed the same survey instrument that was previously utilized in our research detailed in [28], which examined the broader Italian population.” Nevertheless, reference 28 is talking about “Breast Cancer Risk Factors”. 

I therefore recommend checking all references in detail as there might be more mistakes.

2)     In addition, it should/must be clearly presented in the introduction and material and methods if the data published here is "only" a subgroup analysis of already published data.

3)     Regarding my question to the ethical approval of the study by an ethics committee: This information was not satisfactorily updated in the manuscript. Therefore, I must assume, that no ethical approval at an ethics committee was obtained prior to the questionnaire. If so, it should be marked as such in the manuscript. The section on ethics approval is also missing at the end of the manuscript.

4)     The discussion (my main criterion in the last review process) has significantly improved. Nevertheless, it could contain more comparisons to other studies of the existing scientific discourse.
